# Barriers and Facilitators to Artificial Intelligence Implementation in Diabetes Management from Healthcare Workers’ Perspective: A Scoping Review

**DOI:** 10.3390/medicina61081403

**Published:** 2025-08-01

**Authors:** Giovanni Cangelosi, Andrea Conti, Gabriele Caggianelli, Massimiliano Panella, Fabio Petrelli, Stefano Mancin, Matteo Ratti, Alice Masini

**Affiliations:** 1School of Pharmacy, Experimental Medicine and “Stefani Scuri” Public Health Department, University of Camerino, 62032 Camerino, Italy; giovanni01.cangelosi@unicam.it; 2Department of Translational Medicine, Università del Piemonte Orientale, 28100 Novara, Italy; andrea.conti@uniupo.it (A.C.); matteo.ratti@uniupo.it (M.R.); alice.masini@uniupo.it (A.M.); 3Azienda Ospedaliera San Giovanni Addolorata, 00184 Rome, Italy; caggianelligabriele@gmail.com; 4Direzione Medica dei Presidi Ospedalieri, Azienda Ospedaliero-Universitaria di Alessandria, 15121 Alessandria, Italy; 5IRCCS Humanitas Research Hospital, Via Manzoni 56, Rozzano, 20089 Milan, Italy; stefano.mancin@humanitas.it

**Keywords:** diabetes, artificial intelligence, healthcare workers, scoping review, public health

## Abstract

*Background and Objectives:* Diabetes is a global public health challenge, with increasing prevalence worldwide. The implementation of artificial intelligence (AI) in the management of this condition offers potential benefits in improving healthcare outcomes. This study primarily investigates the barriers and facilitators perceived by healthcare professionals in the adoption of AI. Secondarily, by analyzing both quantitative and qualitative data collected, it aims to support the potential development of AI-based programs for diabetes management, with particular focus on a possible bottom-up approach. *Materials and Methods:* A scoping review was conducted following PRISMA-ScR guidelines for reporting and registered in the Open Science Framework (OSF) database. The study selection process was conducted in two phases—title/abstract screening and full-text review—independently by three researchers, with a fourth resolving conflicts. Data were extracted and assessed using Joanna Briggs Institute (JBI) tools. The included studies were synthesized narratively, combining both quantitative and qualitative analyses to ensure methodological rigor and contextual depth. *Results*: The adoption of AI tools in diabetes management is influenced by several barriers, including perceived unsatisfactory clinical performance, high costs, issues related to data security and decision-making transparency, as well as limited training among healthcare workers. Key facilitators include improved clinical efficiency, ease of use, time-saving, and organizational support, which contribute to broader acceptance of the technology. *Conclusions*: The active and continuous involvement of healthcare workers represents a valuable opportunity to develop more effective, reliable, and well-integrated AI solutions in clinical practice. Our findings emphasize the importance of a bottom-up approach and highlight how adequate training and organizational support can help overcome existing barriers, promoting sustainable and equitable innovation aligned with public health priorities.

## 1. Introduction

### 1.1. Prevalence of Diabetes and Social Impact

According to the International Diabetes Federation (IDF), diabetes currently affects approximately 589 million individuals worldwide between the ages of 20 and 79. Among the various forms of diabetes, Type 2 Diabetes (T2D) accounts for the vast majority of cases, with a prevalence estimated between 87% and 91%. Projections for 2050 indicate a substantial increase in global prevalence, expected to reach 853 million cases within the same age range, which will likely be accompanied by a corresponding rise in healthcare costs [1,2,3,4,5]. The majority of individuals with diabetes are obese and physically inactive, particularly within the 45–64 age group (28.9% of males and 32.8% of females) [6,7,8]. In Italy alone, the prevalence of diabetes was estimated at 4 million individuals in 2023 [9,10]. While genetic predisposition and advancing age are recognized contributors to the pathogenesis of numerous chronic diseases such as diabetes [11,12], it is predominantly unhealthy lifestyle behaviors that significantly influence both the onset and progression of these conditions [13,14,15,16,17]. In particular, dietary patterns characterized by excessive caloric intake, a high consumption of refined grains in place of whole grains, and insufficient physical activity constitute the principal modifiable risk factors [18,19,20]. These detrimental habits not only facilitate the development of disease but also exacerbate its clinical course, substantially increasing the risk of complications. Such complications include both peripheral vascular disorders and more complex cardiovascular events, with potentially severe outcomes such as acute myocardial infarction and cerebrovascular accidents (stroke), which are associated with increased morbidity and mortality between in T2D and Type 1 Diabetes (T1D) [21,22,23]. In light of this evidence, the implementation of comprehensive primary and secondary prevention strategies aimed at promoting healthier lifestyles is imperative [24,25,26]. Emphasis should be placed on balanced nutritional intake, caloric moderation, and the adoption of regular physical activity, with the goal of reducing the global burden of disease and improving population health outcomes in T1D and T2D [27,28,29,30].

### 1.2. Use of Devices and Technology in Diabetes Management

At the same time, with the promotion of healthy lifestyle behaviors, diabetes management—particularly for T1D—has long benefited significantly from technological innovation applied in clinical settings, especially with regard to glucose self-monitoring and insulin delivery. Devices such as continuous glucose monitoring (CGM) systems, insulin pumps (IP), and smart multiple daily injection (MDI) systems, often integrated into hybrid closed-loop systems, have become indispensable supports in daily clinical practice [21,22]. They play a critical role in reducing glycemic variability and preventing hypoglycemic episodes, thereby helping to avert major diabetes-related complications. In the context of T2D as well, digital technologies—including self-management applications and telemonitoring tools—are proving valuable, particularly in the personalization of therapeutic interventions and the optimization of clinical outcomes and complication management [31,32,33,34]. In recent years, artificial intelligence (AI) has taken an increasingly prominent role in both diabetes research and clinical practice, owing to its capacity to process and analyze large volumes of clinical and behavioral data efficiently and accurately [35,36,37,38]. Predictive models based on AI techniques such as machine learning (ML) and deep learning (DL) are being developed and implemented to support clinical decision-making, enhance the prediction of complication risks, and personalize treatment pathways [39,40,41]. Applications of AI in diabetes care range from early diagnosis to automated insulin dosing management and the identification of critical glycemic patterns, thereby contributing to the emergence of a new paradigm in precision medicine for diabetes [35,36,37,38,39,40,41]. Nonetheless, the issue of potential barriers and facilitators perceived by healthcare professionals in the effective implementation of these AI-based technologies in routine clinical practice remains largely unexplored—a gap this study aims to address in a bottom-up view, defined as “an approach guided by healthcare workers’ insights and daily experiences” [42].

### 1.3. Study Aims

The primary objective of this study was to investigate the main barriers and facilitators perceived by healthcare professionals involved in the implementation of artificial intelligence (AI) in diabetes management:What are the barriers and facilitators to the use of AI by healthcare professionals in the management of diabetes?

Secondarily, the study aims to explore and support research through the collection of both quantitative and qualitative data to inform the development and implementation of specific AI-based programs for diabetes management, following a bottom-up approach involving healthcare professionals.

Which quantitative and qualitative insights, as perceived by healthcare professionals, can most effectively inform the bottom-up implementation of AI in diabetes care?

## 2. Materials and Methods

### 2.1. Study Design and Registration

A scoping review was conducted to ensure methodological rigor and the relevance of selected studies. This review followed the Preferred Reporting Items for Systematic reviews and Meta-Analyses extension for Scoping Reviews guidelines (PRISMA-ScR) (PRISMA-ScR checklist available in Appendix A) [43]. The protocol for this review was registered in the Open Science Framework database (https://osf.io/xgy2z; accessed on 12 June 2025).

### 2.2. Search Strategy

The search strategy was developed adopting the Population, Concept, Context (PCC) framework (Table 1), without temporal restrictions [44]. The search strategy, updated to 31 January 2025, involved the use of keywords matched using specific Boolean operators such as AND/OR in the databases PubMed Medline, Scopus, CINHAL, and Embase. Search strings are available in Appendix A.

### 2.3. Eligibility Criteria

The inclusion criteria encompassed primary studies published in English without temporal restriction and relevant to the study’s objectives and involving healthcare workers in AI processes. All studies that did not meet the stated inclusion criteria were excluded. The authors nevertheless attempted to include studies in Chinese as well, after evaluating the English-language abstract to ensure it met the inclusion criteria.

### 2.4. Study Selection Process

The study selection process for this review followed a two-phase procedure: an initial screening of titles and abstracts, followed by a detailed evaluation of full-text articles. All potentially relevant articles were imported into the reference manager Ryyan (https://www.rayyan.ai/; accessed on 20 June 2025) for data organization and management. The initial screening was conducted independently and blind by three authors (G.C., A.M., and M.R.), who evaluated titles and abstracts based on their relevance to the study and in accordance with the predefined inclusion criteria. A fourth independent researcher (A.C.) resolved any disagreements at this stage. Following the initial screening, full-text articles meeting the preliminary criteria were independently assessed by the first three researchers, and the fourth still managed conflicts. Any discrepancies were resolved through consensus meetings, with the last researcher (A.C.) acting as an arbitrator to ensure integrity in the selection process. This systematic approach ensured a rigorous and unbiased selection of studies for this review.

### 2.5. Data Extraction and Quality Appraisal

Data extraction from the included studies was organized into key categories, consistent with the methodological framework [43,44]. This structured categorization facilitated both detailed reporting and thorough analysis. The main categories included intervention, outer setting, inner setting, individual characteristics, and implementation process. This structured approach enhanced the clarity and depth of our analysis, aligning with established methodological standards. The extracted data were presented as a narrative summary, organized according to the review’s objectives and supplemented in Table 2. The risk of bias and methodological quality of the included studies were assessed using established guidelines of the JBI framework [43]. Two independent reviewers (A.M. and A.C.) conducted the evaluation to ensure objectivity. Any disagreements were resolved through discussions with a third researcher (M.R.), ensuring that a consensus was reached. The risk of bias and methodological quality of the included studies were evaluated using JBI checklists for qualitative [45] and cross-sectional studies [46] and the MMAT tool for mixed-methods studies [47]. Decisions regarding methodological quality of the studies included were made, independently, by two reviewers, and any disagreements were resolved by discussion. The sum of the points was classified as the percentage of the items present; thus, a score lower than 70% was classified as low-quality, between 70 and 79% of the checklist criteria was classified as medium–high-quality, between 80 and 90% was assigned high-quality, and a score greater than 90% of the criteria was classified as excellent-quality [48]. However, due to the exploratory nature of the present work, no studies were excluded for insufficient quality.

### 2.6. Conceptual and Analytical Framework

The synthesis and presentation of study results followed established guidelines [43,44]. The description of the identified barriers and facilitators is structured according to the Consolidated Framework for Implementation Research (CFIR) [49], a comprehensive theoretical framework widely used to guide implementation research. CFIR comprises five major domains—intervention characteristics, outer setting, inner setting, characteristics of individuals, and implementation process—that offer a systematic approach to understanding factors influencing implementation. In this scoping review, CFIR was used as a guiding structure to map and interpret both qualitative and quantitative data extracted from the selected studies, enabling a comprehensive and theory-informed synthesis of the findings. Key statistical measures, including means (M), standard deviations (SD), and *p*-values, were integral to the analysis. To maintain the integrity of the original studies, statistical significance reporting was preserved as presented in each study. Consistent with scientific conventions, *p*-values of 0.05 or lower were considered statistically significant, ensuring the inclusion of robust and meaningful findings in the review. In addition to the quantitative synthesis, qualitative data were also systematically extracted and analyzed, where applicable, to capture nuanced insights and contextual dimensions of the study findings. The qualitative synthesis followed established principal thematic frameworks in the study included and to further enhance and complete the analysis conducted.

### 2.7. Synthesis of the Results

In this review, while the benefits of meta-analysis are acknowledged, a combined quantitative synthesis was deemed not feasible due to the heterogeneity of the included studies. This variability, characterized by differences in intervention types and methodologies for quantifying relationships between variables, led to inconsistencies in both methodological and statistical approaches. As a result, a detailed narrative synthesis was chosen, following established guidelines for synthesis without meta-analysis (SWiM) [50]. This approach was selected for its effectiveness in transparently and rigorously synthesizing diverse quantitative data, aligning with the PRISMA guidelines [43]. Data synthesis was performed based on the CFIR framework [49]. The CFIR is a well-established conceptual model in implementation science, and it is a comprehensive and standardized meta-theoretical framework. The updated version of the framework is organized into five domains: intervention, outer setting, inner setting, individual characteristics, and implementation process [49]. The CFIR served as a foundational structure for the exploratory assessment of barriers and facilitators to implementing imaging-based, AI-assisted diagnostic decision-making. The comprehensive and adaptable use of the updated CFIR throughout data collection, analysis, and reporting aimed to enhance research efficiency, generate generalizable findings to inform AI implementation practices, and contribute to a robust evidence base for tailoring implementation strategies to overcome key barriers.

## 3. Results

The PRISMA flowchart of the screening process is shown in Figure 1. A total of *n* = 3451 records were retrieved from the databases, and after carefully removing the duplicates (*n* = 593), the researchers (GC, AM, AC, and MR) screened a total of *n* = 3143 for title and abstract. Thirty-two full texts were screened. Finally, a total of *n* = 7 studies, conducted between 2019 and 2024, were included [51,52,53,54,55,56,57].

Studies’ characteristics are shown in Table 2. The population varied from 10 participants to 207, with a range of ages from 40 to 60 years. The involved healthcare professionals were mainly nurses, healthcare assistants, and doctors (i.e., ophthalmologists, diabetologists, endocrinologists, physicians, and general practitioners).

**Table 2 medicina-61-01403-t002:** Summary of the included studies.

Study	Country	Study Design	Setting	Sample Size (N, % Female)	AI Type
Held et al., 2022 [51]	Germany	Qualitative	Primary care	24 (42)	Smartphone-based and AI-supported diagnosis tools for the screening of diabetic retinopathy
Liao et al., 2024 [52]	China	Qualitative	Hospital and community healthcare center	40 (42.5)	AI-assisted system for diabetic retinopathy screening
Petersen et al., 2024 [53]	Denmark	Qualitative	Hospital	18 (61)	AI-assisted system for diabetic retinopathy screening
Romero et al., 2019 [55]	United States	Mixed -methods	Primary care outpatient clinics	83 (N/A)	AI-powered clinical decision support system for identifying diabetes patients at risk of poor glycemic control
Roy et al., 2024 [54]	India	Cross-sectional	Physicians in clinical practice	202 (N/A)	AI-based diabetes diagnostic interventions
Wahlich et al., 2024 [57]	United Kingdom	Qualitative	Hospital and community healthcare center	98 (N/A)	AI-assisted system for diabetic retinopathy screening
Wewetzer et al., 2023 [56]	Germany	Cross-sectional	Primary care	209 (107)	AI-assisted system for diabetic retinopathy screening

AI: artificial intelligence; N: number; N/A: not applicable; %: percentage.

The study conducted by Liao et al. also included healthcare administrative staff and information technology experts [52]. The majority of the included studies were focused on the use of AI for diabetic retinopathy screening [51,52,53,56,57], with only two studies focused on AI as a tool for glycemic control and AI as a general tool for diagnostic intervention [54,55].

Most of the studies adopted a qualitative design (*n* = 4) that used semi-structured interviews [51,52,53,57]. Two quantitative cross-sectional studies were based on surveys [54,56], and finally, one study adopted a mixed-methods design [55].

All the included studies described research conducted in primary care settings, with the exception of the study conducted by Roy et al. [54], which was conducted in a clinic. Notably, two studies were performed inside community healthcare centers [52,57].

### 3.1. Quality Appraisal

Table 3 shows the overall quality of the included studies according to the JBI framework [44]. Complete quality appraisal is available in Appendix A.

Regarding the four qualitative studies, Whalich et al. [57] and Held et al. [51] were rated as medium-quality due to the absence of statements addressing the researchers’ cultural positions and the mutual influence between the researchers and the research process. The studies by Liao et al. [52] and Peterson et al. [53] were well designed.

The two studies with a cross-sectional design presented as low-quality due to the lack of information during the description of the inclusion and exclusion criteria [54] and the setting of the study, while the study by Wewetzer et al. [56] was reported as medium-quality owing to the lack of managing confounding factors.

Finally, the mixed-methods study performed by Romero et al. [55] all was well designed and conducted, with a high-quality rating.

### 3.2. Barriers and Facilitators Identified According to the CFIR Framework

The following sections present findings according to the CFIR framework domains, and Figure 2 illustrates the main identified barriers and facilitators. Comprehensive data extraction is shown in Appendix A.

#### 3.2.1. Individuals Domain

The principal barrier is the negative personal attitude towards AI systems, with associated skepticism [51], incompetence in understanding the AI reasoning mechanisms [52], or forgetfulness to visit the patient [53].

The facilitators belonging to this domain are engagement of hospital administrators or department heads [52], positive attitude towards the future of AI technology [57], perception of increased care coordination [55], or perception of AI as a support [57].

#### 3.2.2. Intervention Domain

The relevant barrier under this domain is the unsatisfactory clinical performance of the AI system. This aspect is related to image recognition, time (duration of examination and latency of results), validity (AI system may miss some retinopathy changes), and uncertainty about the accuracy and trustworthiness of AI outputs [51,52,53,54]. Also, Romero et al. [55] reported as a barrier the high false positive rate in patient risk classification, whereas the studies of Wewetzer et al. [56] and Wahlich et al. [57] highlighted diagnostic limitations of AI systems (they may not detect other conditions besides the one for which they have been designed for, leading to incomplete diagnosis). Finally, doubts about reliability and accuracy may negatively impact physician confidence in the system [56,57]. Another common barrier is related to the financial burden of AI software/high acquisition costs [52,53,56].

Another fundamental barrier to consider is the concern about data security, liability, and how the system made the decision (black box problem) [54,55,57]. Romero et al. [55] and Liao et al. [52] also reported the poorly tailored reporting of AI systems.

The identified facilitators are an improved clinical efficiency [52,54,55], easiness of use [52,54], and the time sparing effect, both for the patient and for the physician [53,56,57]. Only one study reported a financial facilitator [51].

#### 3.2.3. Implementation Process Domain

The principal barriers are the lack of feedback in incorporation of the system [52], the increased workload due to extra steps outside the routine workflow [55], and the impact of a grading workload [57].

The only facilitator identified in the implementation process is the successful active engagement of the clinicians [52].

#### 3.2.4. Inner Setting Domain

The principal barrier related to the inner setting is the lack of adequate training or limited knowledge and training on the AI system [52,54,55]. Also, the lack of integration of the AI system into the hospital/facility information system is another relevant concern that acts as a barrier [54,55,56].

The only inner setting facilitator identified under this domain is the ease of integration, with the associated simple installation [51,52].

#### 3.2.5. Outer Setting Domain

For the outer setting domain, the crucial emerging barrier is the lack of a collaborative network between primary, secondary, and tertiary hospitals, which includes the related tensions between GPs and specialists with the concern of a lower referral rate [51,52]. Also, the fear of job displacement or changes in clinical autonomy is another concern that is reported as a barrier [54]. A study also reported that insufficient reimbursement by health care systems may act as a significant barrier [56].

The facilitators reported that can be classified under this domain are the development of national guidelines related to AI [52], the strengthening of the primary doctor filter function, and an associated closer relationship between patient and GP [51].

## 4. Discussion

The implementation of AI systems in healthcare represents one of the most significant challenges of the coming decade, not only for daily clinical practice but also for public health systems as a whole. The findings of this study, interpreted through the CFIR framework [49], clearly show that the barriers and facilitators to AI adoption operate on multiple levels, reflecting organizational, technological, individual, and systemic dynamics that influence full implementation and value realization.

### 4.1. Innovation, Effectiveness, and Trust in Technology

The most critical issue that emerged relates to the perceived unsatisfactory clinical performance—an aspect with profound implications for public health [58,59]. If AI systems are unable to ensure adequate levels of sensitivity and specificity—or if their reliability is perceived as uncertain—they risk undermining the quality of care and increasing the chances of missed or inappropriate diagnoses [60]. In the context of population screening (such as diabetic retinopathy), a high number of false positives or false negatives could either overload the system or falsely reassure patients. This confirms what has already been observed in international studies, which emphasize the need for rigorous clinical validation before the large-scale deployment of AI systems [61,62]. Furthermore, the lack of transparency in AI’s decision-making processes undermines the trust of healthcare professionals and patients, posing an ethical and regulatory challenge essential for the sustainable development of the technology [60,63]. This issue is currently the focus of attention within the European Health Data Space and the EU Artificial Intelligence Act, which introduces strict requirements for the “explainability” and traceability of collected data [64]. These concerns are rooted primarily in the CFIR domain of intervention, where the perceived evidence strength, complexity, and relative advantage of the AI tools are critical to their adoption. Additionally, elements from the individuals domain emerge, particularly regarding trust and acceptance by healthcare professionals.

### 4.2. Equity and Sustainability: AI Costs, Access, and Integration

The barrier represented by the costs of acquiring and managing AI can also be interpreted in terms of equity in access to healthcare innovations [65]. This economic barrier was identified within the CFIR domain of intervention characteristics, highlighting the perceived complexity and resource intensity associated with adopting AI-based tools. Mainly in publicly funded universal healthcare systems, the adoption of expensive technologies risks creating territorial inequalities, where larger or better-funded facilities can afford to implement AI, while peripheral or resource-limited ones risk being effectively excluded [66]. This issue was already highlighted by the World Health Organization (WHO) in the 2021 report “Ethics and Governance of Artificial Intelligence for Health”, which urges careful evaluation of AI’s redistributive impact in public health contexts [67]. Adding to this is the poor integration with existing information systems, which are often not designed to accommodate supplementary technologies, thereby hindering true interoperability. Without full connectivity between AI, electronic health records, and healthcare management systems, AI risks remaining an isolated technology, incapable of delivering real value to patient care [68,69]. These considerations also align with the principles of the United Nations Sustainable Development Goals (SDGs), particularly those promoting health, innovation, and equity in access to care [70].

### 4.3. Healthcare System and Multilevel Governance

In terms of governance and shared decision-making, it becomes clear that the lack of coordination between levels of care (e.g., primary care, specialists, hospitals) and potential communication difficulties among professionals represent critical barriers to AI implementation. This issue is part of a broader challenge in healthcare governance, which calls for dynamic and modern models of both vertical and horizontal service integration [71,72,73]. Technology alone does not solve these problems; on the contrary, it can amplify them if introduced without a clear and shared organizational framework. The concern, particularly regarding the potential loss of clinical autonomy—clearly emerging from the data reviewed—is a key issue for the “social” legitimacy of digital innovations in healthcare [74,75]. According to Greenhalgh et al. (2017), the adoption of new technologies requires a co-construction of change, in which professionals are actively involved not only in the use but also in the design and evaluation of the tools to be implemented in clinical practice [76]. These barriers align with the CFIR domains of *inner setting* and *implementation process*, emphasizing organizational coordination gaps and limited stakeholder engagement in technology adoption.

### 4.4. Training, Digital Literacy, and Empowerment

These issues fall within the CFIR domain of characteristics of individuals, particularly focusing on knowledge, self-efficacy, and the need for ongoing professional development to support effective AI adoption. From an internal perspective, the lack of adequate training on AI tools is a systemic barrier that must be urgently addressed—especially if the goal is to adopt AI-based care systems across multiple levels of healthcare delivery [77]. Structured pathways for continuous education are therefore essential, integrating “AI literacy” into university curriculum, ongoing professional development programs, and career advancement tracks. This ensures they do not become mere executors but rather informed actors who use technology as an extension of their clinical skill expertise [78,79].

### 4.5. Opportunities for Public Health

Despite the critical issues that have emerged, the identified facilitators show significant transformative potential. Time savings, increased clinical efficiency, the perception of greater care coordination and positive acceptance by some operators are valuable elements for the success of public policies that aim at the equitable and sustainable digitalization of healthcare [80,81,82].

The presence of possible national and international guidelines to support health professionals is perceived as an enabling element, and in this scenario, AI could represent a turning point if integrated with global implementation strategies, clinical audits, and impact assessment [64]. It is desirable that an international public control room is created to guarantee the implementation of AI, capable of providing technical standards, ethical assessments, and support to local decision makers and organizers of health services in general that meet all the quality standards necessary for the use of AI in clinical practice [67,83]. These facilitators reflect the CFIR domains of *outer setting* and *implementation process*, particularly highlighting the role of policy support, external incentives, and structured strategies to guide effective and equitable AI integration.

### 4.6. Barriers and Facilitators in a Bottom-Up Perspective

The introduction of AI and other technological tools into clinical settings—particularly for the management of chronic diseases—represents a significant opportunity to rethink traditional models of care and transition toward a more predictive, proactive, and personalized approach [84,85,86,87,88,89]. Chronic conditions such as diabetes, hypertension, cardiovascular, and respiratory diseases represent a substantial burden for public healthcare systems and require coordinated, continuous, and patient-centered management strategies [90,91]. In this context, AI can serve as a catalyst for innovation, provided that it is embedded in a well-structured and responsive clinical and organizational ecosystem [92]. Building on the CFIR framework, our analysis identified key barriers and facilitators across its five domains: characteristics of the intervention, outer setting, inner setting, individual characteristics, and implementation process. These domains provided the foundation for a structured and theory-informed classification of the data, while the discussion reframed these results in light of a bottom-up perspective—that is, how professionals working within healthcare systems perceive, experience, and respond to the integration of AI in their clinical practice [49,93,94]. From a practical standpoint, AI-based systems—such as those used for diabetic retinopathy screening or cardiovascular risk stratification [95,96]—can support the early detection of complications, thereby reducing the need for hospital-based interventions and enabling more timely, preventive care. This dual benefit supports both patients, who receive appropriate interventions sooner, and healthcare systems, which benefit from improved outcomes and resource efficiency [97]. However, integrating AI into routine clinical practice necessitates parallel transformations in organizational routines, workforce training, and digital competencies among healthcare professionals [98]. Rather than framing AI as a substitute for human expertise, it is essential to foster a collaborative model in which technology enhances the clinical judgment and skills of professionals. This reframing can help overcome cultural resistance to adoption, often rooted in concerns about professional identity and loss of autonomy [99,100]. Among the barriers, frontline professionals reported challenges such as poor interoperability of AI with existing electronic health records, lack of shared implementation strategies, and insufficient digital literacy. These aspects reflect issues related to the CFIR domains of intervention characteristics, inner setting, and outer setting. Additionally, fragmented care coordination and lack of shared governance structures were seen as limiting factors in the effective deployment of AI across healthcare levels [101,102]. Conversely, several facilitators emerged from a bottom-up viewpoint, highlighting how AI adoption can be positively influenced by perceived improvements in workflow efficiency, time savings, and greater integration between primary and specialty care. Professionals also reported a willingness to engage with new tools in general technology devices when training is adequate and when they are actively involved in implementation decisions in clinical and social decision [103,104,105].

### 4.7. A Public Health View on AI in Clinical Practice

For chronic disease management, where continuity and coordination of care are essential, AI systems can support transitions between care settings, enhance remote monitoring, and offer predictive insights that guide therapeutic decisions [106,107,108]. However, without appropriate regulation and inclusive planning, the risk of exacerbating health inequities remains high—particularly in underserved or digitally marginalized populations [109,110]. To prevent such disparities, healthcare professionals must be empowered not only as end users but also as digital mediators and educators, helping to guide patients and caregivers in the effective and ethical use of AI technologies [111,112,113,114,115]. Their proximity to patients and deep understanding of local care pathways position them as crucial actors in fostering responsible innovation. Finally, the development of AI for chronic care must be embedded in a broader cultural and ethical framework that views data as a collective asset and public health as the primary driver for technological development. This shift is necessary to ensure that data-driven medicine evolves in a fair, sustainable, and inclusive direction [116,117].

### 4.8. Strengths and Limitations

The study’s strengths include its solid methodological framework, facilitated by the adoption of the JBI methodology, the PRISMA-ScR guidelines for reporting [43,44], and the registration of the protocol on the OSF database, elements that ensure transparency and international rigor. The use of the CFIR framework [49] allowed for an in-depth analysis of the factors influencing AI adoption by healthcare workers, facilitating a coherent and useful classification from a public health perspective. However, some limitations should be considered. The exploratory nature of the scoping review did not allow for a quantitative synthesis of data due to the methodological heterogeneity of the included studies and the cohorts considered. Additionally, many findings are based on subjective perceptions of the involved workers in the studies included, which requires caution in interpreting the collected data and highlights the need for future integration with objective data and evidence-driven research. For these elements, future studies aimed at addressing these gaps are required. In particular, it is suggested to develop longitudinal research and randomized studies that evaluate not only perceptions but also the real effectiveness and efficiency of AI in managing diabetes and chronic diseases in general from a healthcare worker’s point of view. This approach would help build more comprehensive and useful evidence for the sustainable implementation of technological innovation in public health perspectives.

## 5. Conclusions

The adoption of AI in chronic care, particularly in diabetes management, is shaped by a dynamic interplay of perceived barriers and facilitators among healthcare professionals. Their active and continuous involvement represents a key opportunity to develop more effective, reliable, and context-aware AI solutions that are better integrated into everyday clinical workflows. Promoting targeted training programs and sustained organizational support for healthcare workers involved in the complex management of diabetes may help overcome current challenges, advancing more equitable and sustainable innovation. Crucially, these findings point toward the value of a bottom-up approach—one that prioritizes the perspectives and practical needs of frontline professionals—as a promising pathway to support the successful implementation of AI in clinical practice. Framing AI adoption within a broader public health perspective, attentive to systemic readiness and social equity, can guide more inclusive strategies that align innovation with real-world healthcare priorities.

## Figures and Tables

**Figure 1 medicina-61-01403-f001:**
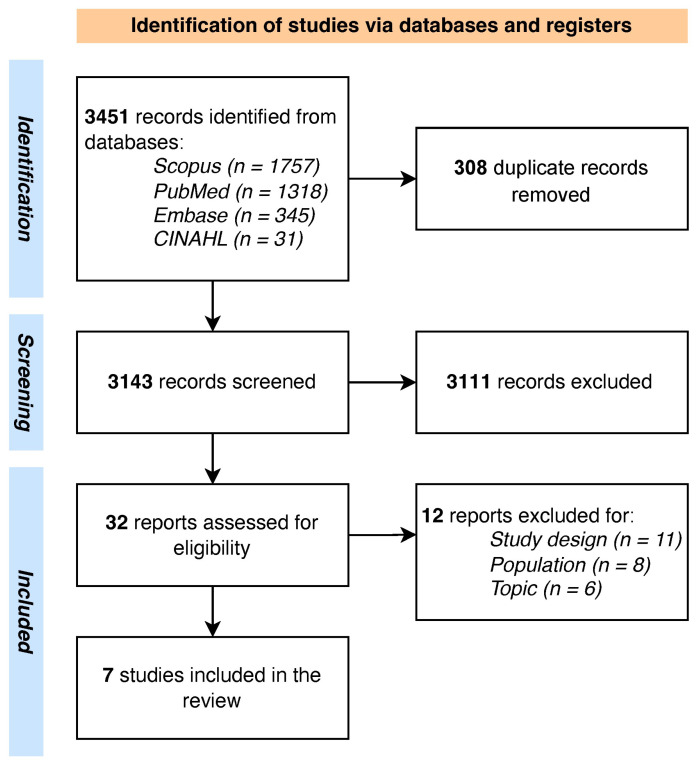
PRISMA flowchart.

**Figure 2 medicina-61-01403-f002:**
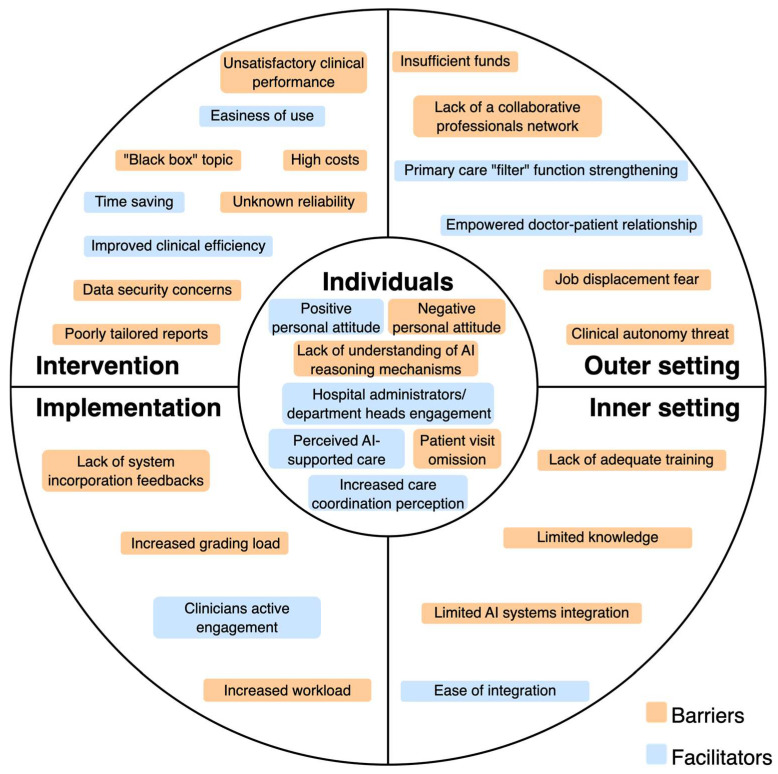
Barriers and facilitators of AI implementation for diabetes management reported by healthcare professionals.

**Table 1 medicina-61-01403-t001:** Inclusion and exclusion criteria, described according to the PCC framework.

Parameter	Inclusion Criteria	Exclusion Criteria
Population	Studies involving healthcare professionals (principally doctors, nurses, specialists, technicians) who manage diabetes with AI.	Studies that do not involve healthcare workers.
Concept	Studies exploring the adoption and implementation of AI in managing diabetes, such as monitoring systems, diagnostics, predictive therapy, and personalized patient management. AI technologies extended in ML or DL use.	Studies that address AI in non-healthcare contexts or those unrelated to managing diabetes; technological interventions that do not use AI, ML or DL.
Context	Barriers and obstacles perceived from healthcare workers in adopting AI (e.g., technological difficulties, cultural challenges, insufficient training, resistance to change). Facilitators and enabling factors from healthcare workers in terms of AI adoption (e.g., organizational support, training, technology accessibility, evidence of effectiveness).	Studies that do not explore barriers or facilitators in AI adoption by healthcare workers; research that only addresses clinical outcomes of diabetes treatment without focusing on perception, implementation science, and attitude.

AI: artificial intelligence; ML: machine learning; DL: deep learning.

**Table 3 medicina-61-01403-t003:** Results of quality appraisal.

Study	Checklist	Overall Quality
Held et al., 2022 [51]	JBI for qualitative studies	Medium
Liao et al., 2024 [52]	JBI for qualitative studies	Excellent
Petersen et al., 2024 [53]	JBI for qualitative studies	High
Romero et al., 2019 [55]	MMAT	High
Roy et al., 2024 [54]	JBI for analytical cross-sectional studies	Low
Wahlich et al., 2024 [57]	JBI for qualitative studies	Medium
Wewetzer et al., 2023 [56]	JBI for analytical cross-sectional studies	Medium

MMAT: Mixed-Methods Appraisal Tool; JBI: Joanna Briggs Institute.

## Data Availability

Data is contained within the article or Appendix A.

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
