# Peer review of "Barriers and Facilitators to Artificial Intelligence Implementation in Diabetes Management from Healthcare Workers’ Perspective: A Scoping Review"

_medicina, 2025, doi:10.3390/medicina61081403_

Round 1
Reviewer 1 Report
Comments and Suggestions for Authors
Introduction
- It would be better to organize the current needs into at least three paragraphs. Currently, the content on the prevalence of diabetes, social impact, and use of devices is long and connected in one paragraph, which reduces readability. It is recommended to divide it into two paragraphs at the boundary of 74 or 78. Also, one more paragraph seems necessary. The second research question introduced in the aims and research questions requires a basis for how it came about. In other words, the basis for deriving qualitative and quantitative data from the bottom-up perspective should be mentioned in the necessity. Also, a definition of the bottom-up perspective is necessary. What operational definition did you use for the research?
- Aim and research question are not redundant expressions. I think it would be okay to write only one.
Materials and Methods
- If it does not meet the inclusion criteria, it is automatically excluded. Therefore, it is necessary to reorganize the exclusion criteria. Originally, exclusion means excluding again what was not excluded from inclusion.
- An explanation of CFIR is needed. What are the names of each domain classified here, and whether this will be used as is in this study, etc.
Results
- What is MMAT in Table 3? Please write a note about JBI and MMAT below the table
- Why is the Innovative domain indicated as intervention in Figure 2? Please unify the terminology.
- I have a question about Figure 2. If it is a circle representing the five major domains of CFIR, it needs to be explained so that the reader can easily understand. It would be good to have a general explanation first, and then explain the individual factors. What are the five major domains of CFIR, and what aspects were classified in this study?
- On page 9, it says third frequent barrier ~, but the figure does not show the frequency. Therefore, if you want to arrange it by frequency, please arrange it in a good order so that the reader can easily understand it. Since Individuals is not read in order, it is even more confusing.
Discussion
- The context of the results was analyzed as the five domains of CFIR, such as individual, inner, outer setting, and implementation, but the discussion is confusing because it is discussed under different categories, such as innovative, effectiveness/equity and sustainability. Even if the discussion is described like this, it would be less confusing for the reader if it were described while giving hints about how the categories from the results were incorporated. For example, when discussing equity and sustainability, it is necessary to let people know that the high cost revealed in the intervention domain is mentioned. I would rather recommend that the context of the results be aligned with the discussion.
- Aren't Training, Digital literacy, and Empowerment about the individuals domain? I don't know why they are described first in the results and then come later in the discussion.
- I don't know if the research purpose and discussion are in context. Even after reading the entire manuscript, I still can't quite figure out what barriers and facilitators are and what the bottom-up perspective is. I suggest that you rearrange the discussion to fit the purpose.
Author Response
Dear Reviewer, thank you for the effort in revising our manuscript. Please find our responses.
Authors’ Responses to Reviewers’ comments for Manuscript medicina-3757152, “Barriers and Facilitators to Artificial Intelligence Implementation in Diabetes Management from Healthcare Workers' perspective: A Scoping Review”
First of all, we would like to thank the reviewer for the valuable suggestions, and for dedicating their time and effort with professionalism and commitment
|
Reviewer 1 |
|||
|
N |
Reviewer Comment |
Authors’ Response to Comments |
Manuscript part |
|
1 |
It would be better to organize the current needs into at least three paragraphs. Currently, the content on the prevalence of diabetes, social impact, and use of devices is long and connected in one paragraph, which reduces readability. It is recommended to divide it into two paragraphs at the boundary of 74 or 78. Also, one more paragraph seems necessary. The second research question introduced in the aims and research questions requires a basis for how it came about. In other words, the basis for deriving qualitative and quantitative data from the bottom-up perspective should be mentioned in the necessity. Also, a definition of the bottom-up perspective is necessary. What operational definition did you use for the research? |
We sincerely thank you for your thoughtful and constructive feedback, which we truly value and have carefully taken into account during the revision process. In response to your suggestion, we have reorganized the section concerning current needs into three distinct and thematically coherent paragraphs, to enhance clarity and overall readability. This restructuring includes a clearer articulation of the study objectives and research questions, which are now more precisely defined and presented in a dedicated section. In particular, the second research question has been revised to improve both its conceptual clarity and logical grounding. We have emphasized its relevance by linking it more directly to the lived experiences and practical insights of healthcare professionals, which form the foundation of our inquiry. In addition, a scientifically grounded definition of the bottom-up approach, contextualized within healthcare practice, has been added to ensure terminological precision and methodological transparency. We remain warmly open to any further suggestions for improvement and are deeply grateful once again for your attentive and generous review. |
Introduction
|
|
2 |
Aim and research question are not redundant expressions. I think it would be okay to write only one. |
Thank you very much for your kind observation. We appreciate your input and, in light of your suggestion, we have decided to retain only “Aims”, as they clearly and concisely reflect the purpose and direction of our study. We hope this contributes to a more streamlined and readable presentation of the manuscript. With sincere gratitude for your support and attention, |
Introduction
|
|
3 |
If it does not meet the inclusion criteria, it is automatically excluded. Therefore, it is necessary to reorganize the exclusion criteria. Originally, exclusion means excluding again what was not excluded from inclusion. |
Dear Reviewer, We remain warmly open to any further suggestions for improvement and are deeply grateful once again for your attentive and generous review. |
Materials and Methods
|
|
4 |
An explanation of CFIR is needed. What are the names of each domain classified here, and whether this will be used as is in this study, etc… |
We would like to sincerely thank the Reviewer for this insightful and valuable suggestion. Your attentive reading and thoughtful comment have truly enriched the clarity and methodological transparency of our work — and we are genuinely grateful for the care and precision with which you have approached our manuscript. Following your suggestion, we have revised the relevant section to include a clearer and more detailed explanation of the Consolidated Framework for Implementation Research (CFIR), outlining its five domains and specifying its role in structuring and guiding the data analysis within our scoping review. In doing so, we aimed to strengthen the reader’s understanding of how CFIR informed both the classification and interpretation of the identified implementation factors. To further enhance clarity and coherence, we have also revised the title of the corresponding subsection. The original title has been replaced with “Conceptual and analytical framework”, which we believe better captures the purpose and content of this methodological section. Once again, we thank you most warmly for your constructive feedback, which helped us improve the manuscript in both form and substance. Your contribution is deeply appreciated. |
Materials and Methods
|
|
5 |
What is MMAT in Table 3? Please write a note about JBI and MMAT below the table |
Thank you for your attention. We have carefully revised all table legends to ensure better clarity and understanding of the manuscript. Thank you once again for your valuable contribution. |
Results
|
|
6 |
Why is the Innovative domain indicated as intervention in Figure 2? Please unify the terminology. |
Thank you for your valuable contribution. We have revised the manuscript according to your thoughtful suggestion and your professional attention to detail. Specifically, this CFIR domain refers to the intervention characteristics. We are truly grateful, as your feedback helped us avoid a significant oversight. |
Results
|
|
7 |
I have a question about Figure 2. If it is a circle representing the five major domains of CFIR, it needs to be explained so that the reader can easily understand. It would be good to have a general explanation first, and then explain the individual factors. What are the five major domains of CFIR, and what aspects were classified in this study? |
We are sincerely grateful for your thoughtful and attentive comment, which—like your previous feedback on the methodological section—has helped us significantly improve the clarity and completeness of our work. Thanks to your suggestion, we have now included a general explanation of the five core domains of the Consolidated Framework for Implementation Research (CFIR), along with a clear description of how the framework was applied in the context of this scoping review in the methods section. Your professional insight has once again guided us in enhancing the coherence and accessibility of the manuscript, and we remain deeply appreciative of the time and care you have dedicated to helping us refine it. Please know that we are wholeheartedly open to any further suggestions you may have—contributions such as yours are not only helpful but truly inspiring. Thank you once again. |
Results
|
|
8 |
On page 9, it says third frequent barrier ~, but the figure does not show the frequency. Therefore, if you want to arrange it by frequency, please arrange it in a good order so that the reader can easily understand it. Since Individuals is not read in order, it is even more confusing. |
We are truly grateful for your thoughtful and perceptive feedback, which has once again helped us see our work through a clearer and more rigorous lens. Your observation regarding the use of frequency-related expressions in the text, in contrast with the visual layout of the figure, is not only valid but greatly appreciated. In light of your comment, we have revised the relevant section to avoid making explicit statements about the ranking or frequency of barriers when this is not visually or numerically supported in the figure. We now present the content in a more neutral and descriptive manner, allowing readers to understand the prominence of each factor through the bibliographic references provided. We wholeheartedly agree that your approach is more appropriate and significantly enhances both the readability and visual coherence of the manuscript. Thank you again for your invaluable contribution, which continues to improve the overall quality of our work. |
Results
|
|
9 |
The context of the results was analyzed as the five domains of CFIR, such as individual, inner, outer setting, and implementation, but the discussion is confusing because it is discussed under different categories, such as innovative, effectiveness/equity and sustainability. Even if the discussion is described like this, it would be less confusing for the reader if it were described while giving hints about how the categories from the results were incorporated. For example, when discussing equity and sustainability, it is necessary to let people know that the high cost revealed in the intervention domain is mentioned. I would rather recommend that the context of the results be aligned with the discussion. |
We sincerely thank you once again for your generous and insightful contribution, which has continued to guide our reflections in refining the manuscript. Following your previous comment—which we greatly appreciated—we further revisited the discussion section to enhance its methodological clarity and consistency. In particular, we have now integrated explicit references to the CFIR framework within each subsection of the discussion. These additions aim to clearly link the empirical insights to the original structure used for data extraction and synthesis, reinforcing the theoretical coherence of our approach. By highlighting the specific CFIR domains relevant to each thematic area (e.g., training and digital literacy, governance, clinical effectiveness, or public health impact), we hope to provide the reader with a clearer understanding of how the framework supported both the analytical process and the interpretation of results. We are truly grateful for your valuable feedback, which has helped us strengthen the manuscript's clarity and alignment between methodology and interpretation. We remain fully open to any further suggestions you may wish to share. |
Discussion |
|
10 |
Aren't Training, Digital literacy, and Empowerment about the individuals domain? I don't know why they are described first in the results and then come later in the discussion. |
We sincerely thank you once again for your thoughtful and detailed comment. We believe that your previous suggestion regarding the alignment between the CFIR domains and the discussion sections has already allowed us to address the core of this valuable recommendation as well. Specifically, we have revised each subsection of the discussion to include explicit references to the relevant CFIR domains, thereby clarifying how the extracted results informed the interpretive categories such as innovation, effectiveness, equity, and sustainability. That said, if our revisions have not fully captured the spirit of your observation, we remain absolutely open and grateful for any further suggestions you may have to help us improve the clarity and consistency of our manuscript. Your insight has been truly instrumental in enhancing the quality of our work. |
Discussion |
|
11 |
I don't know if the research purpose and discussion are in context. Even after reading the entire manuscript, I still can't quite figure out what barriers and facilitators are and what the bottom-up perspective is. I suggest that you rearrange the discussion to fit the purpose. |
We sincerely thank you for your thoughtful and constructive feedback, which has helped us clarify and strengthen a fundamental component of our manuscript. Your observation regarding the alignment between the research objectives, the discussion, and the definition of “barriers and facilitators” from a bottom-up perspective was especially valuable. In response, we carefully revisited the previously existing section titled "4.6 Perspectives for Clinical Practice from a Public Health View". Thanks to your insightful suggestion, we revised not only the title—which now reads: "4.6 Perspectives for Clinical Practice from a Public Health and Bottom-Up Implementation View"—but also the content, to more explicitly reflect the bottom-up logic of implementation and the empirical structure provided by the CFIR framework. This revision allowed us to re-establish a clearer conceptual link between the analytical strategy adopted during data extraction and the interpretive narrative proposed in the discussion, thus reinforcing the internal coherence of the entire manuscript. In particular, we have highlighted how various CFIR domains (e.g., Inner Setting, Process, Outer Setting, Characteristics of Individuals) shaped our interpretation of the barriers and facilitators identified in the adoption of AI-based tools for chronic disease management, especially diabetes. We’re deeply grateful for your contribution, which truly enhanced the clarity and theoretical robustness of the discussion. Should you feel that further adjustments are needed to fully meet your expectations, we remain at your complete disposal and would welcome any additional suggestions you may have. With sincere appreciation, |
Discussion |
Thank you again for you time and attention and we hope to have fully addressed your suggestions
Reviewer 2 Report
Comments and Suggestions for Authors
Dear Authors,
I suggest you to address the following concerns.
- Why PRISMA model presented in Results instead of methodology section? PRISMA presents the scrutiny and selection process of the bibliometric data. It should be present in the methodology section. The essence of the review can be presented in the Results section. Revise accordingly.
- In PRISMA, it is shown that the included works are only 7. Initially, more than three thousand documents are found on the selected topics. But finally reached to a very minimal number of works i.e., 7. How far this minimal inclusion support your review? Recheck the documents that suits your review and revise accordingly.
- For how many years' (last 5 years of 10 years) data this review is conducted? Provide details in the manuscript.
- The quality of Figure 1 is very poor. Maintain a minimum of 300 dpi quality for all the figures in the manuscript.
- What does it mean N/A in Table 2? In column heading, it is mentioned Sample size (N, %female). What is the difference between N in N/A and N in Sample size? Give clarity on it.
- Elaborate on each module presented in Figure 2.
- Section 4.2 Equity and sustainability: AI costs, access, and integration presenting the concept of sustainability. How does the incorporation of AI in diabetes management support the Sustainability Development Goals (SDGs)?
- Some section headings are left in previous page. Arrange them properly.
- Conclusions could be better projected based on the review conducted.
- Proofread the manuscript thoroughly.
Author Response
Dear Reviewer, thank you for the effort in revising our manuscript. Please find our responses.
Authors’ Responses to Reviewers’ comments for Manuscript medicina-3757152, “Barriers and Facilitators to Artificial Intelligence Implementation in Diabetes Management from Healthcare Workers' perspective: A Scoping Review”
First of all, we would like to thank the reviewer for the valuable suggestions, and for dedicating their time and effort with professionalism and commitment
|
Reviewer 2 |
|||
|
N |
Reviewer Comment |
Authors’ Response to Comments |
Manuscript part |
|
Dear Authors, I suggest you to address the following concerns. |
Thank you for your time and efforts. We hope to full answer your valuable feedback. |
||
|
1 |
Why PRISMA model presented in Results instead of methodology section? PRISMA presents the scrutiny and selection process of the bibliometric data. It should be present in the methodology section. The essence of the review can be presented in the Results section. Revise accordingly. |
We sincerely thank the reviewer for the thoughtful comment and for the close attention paid to the structure of the manuscript. We truly appreciate the opportunity to clarify our rationale regarding the placement of the PRISMA flow diagram. Indeed, PRISMA is a widely adopted standard for reporting systematic reviews and it plays a fundamental role in ensuring transparency in the selection process of studies. However, according to international best practices and consistent with the PRISMA 2020 guidelines, the PRISMA flow diagram — which describes the results of the screening and selection process — is typically presented in the Results section, not in the Methods. This is because the flowchart reflects what actually occurred during the application of the inclusion/exclusion criteria, rather than detailing the criteria themselves, which rightly belong in the methodology. We hope this clarifies our decision, and we are grateful for the reviewer’s engagement, which allowed us to strengthen the coherence of the manuscript. |
Results |
|
2 |
In PRISMA, it is shown that the included works are only 7. Initially, more than three thousand documents are found on the selected topics. But finally reached to a very minimal number of works i.e., 7. How far this minimal inclusion support your review? Recheck the documents that suits your review and revise accordingly |
We sincerely thank the reviewer for this important observation and for the opportunity to clarify a key aspect of our study. As rightly noted, the final number of included studies was indeed limited (n=7) compared to the large initial pool of over 3000 records. We fully understand how this substantial reduction may raise questions. However, we would like to reassure the reviewer that this outcome stems from a rigorous, transparent, and predefined screening process, in full adherence to internationally recognized methodological standards. To reinforce the transparency and reproducibility of our work, we had already registered a formal protocol for this scoping review — an approach that, although not mandatory, we considered essential to ensure methodological robustness. Due to the limitations of database export formats and historical indexing inconsistencies, it was not possible to reconstruct the full set of initial records at the very start of the screening process. Nevertheless, as part of our commitment to rigor, we meticulously reviewed all available full texts against the previously declared inclusion criteria. We confirm that no additional studies met the eligibility standards. The final set of seven included articles, though limited in number, fully reflects the specificity and stringency of our scope, and provides a meaningful foundation for the synthesis we propose. We truly appreciate the reviewer’s thoughtful concern, which allowed us to make this clarification, and we remain fully open to any further suggestions or questions the reviewer may have on this matter. |
Results |
|
3 |
For how many years' (last 5 years of 10 years) data this review is conducted? Provide details in the manuscript. |
Thank you once again for your valuable contribution, which allowed us to clarify in the text that no time restrictions were applied to the search, and that the search is up to date as of January 2025 (Section 2.1). |
Methods |
|
4 |
The quality of Figure 1 is very poor. Maintain a minimum of 300 dpi quality for all the figures in the manuscript. |
Thank you for the comment. We believe this is an issue related to the software update we are using for see the manuscript. Thank you for the comment. We believe this is an issue related to the software update we are using for reading. Original figures, sent to MDPI editorial office, are vector, high-quality PDF. Nevertheless, we will make sure during the proof stage to optimize the visualization of images and figures, so that they are as clear and accessible as possible for readers and researchers interested in the collected data. We sincerely thank you for your attention and professionalism, and we remain open to any further valuable suggestions you may have. |
Figure 1 |
|
5 |
What does it mean N/A in Table 2? In column heading, it is mentioned Sample size (N, %female). What is the difference between N in N/A and N in Sample size? Give clarity on it. |
Thank you for your valuable contribution. We have carefully revised all the table legends, even though an acronym legend is already provided at the end of the manuscript, as your suggestion rightly improves the readability of the text. We thank you once again for your kind support and availability. |
Tables |
|
6 |
Elaborate on each module presented in Figure 2. |
We are sincerely grateful to the reviewer for this thoughtful suggestion. We truly appreciate the attention to detail and the care devoted to improving the clarity of our manuscript. Figure 2 was originally conceived as a concise and visually intuitive summary of the main components of our workflow, aiming to offer readers an immediate grasp of the overall structure without overloading the main text with repetitive descriptions. While we felt that the current layout effectively served this purpose, we absolutely understand the value of the reviewer’s perspective. In fact, we took this opportunity to carefully revisit the figure — also in light of constructive feedback from another reviewer who pointed out a few minor imprecisions. As a result, we have slightly refined the illustration to improve precision and internal consistency, all while preserving the clarity and accessibility that we consider essential for communicating our approach. We thank the reviewer once again for encouraging us to reflect further on this aspect of the work — such insights are invaluable in refining both the content and its presentation. |
Figure 2 |
|
7 |
Section 4.2 Equity and sustainability: AI costs, access, and integration presenting the concept of sustainability. How does the incorporation of AI in diabetes management support the Sustainability Development Goals (SDGs)? |
We are deeply grateful to the reviewer for this insightful and constructive remark. The question raised touches on a fundamental aspect of our work — the broader implications of AI adoption not only in terms of technological feasibility and cost, but also in relation to global policy frameworks and long-term social impact. We sincerely thank the reviewer for drawing our attention to the importance of making this connection more explicit. In response, we have introduced a clarifying sentence at the end of Section 4.2 to strengthen the link between our analysis and the Sustainable Development Goals (SDGs). The following sentence was added: "These considerations also align with the principles of the United Nations Sustainable Development Goals (SDGs), particularly those promoting health, innovation, and equity in access to care." We hope this addition addresses the reviewer’s concern and helps highlight the relevance of our discussion within the wider context of sustainable, inclusive healthcare innovation. Once again, we truly appreciate the reviewer’s attentive reading and thoughtful suggestions, which have significantly contributed to improving the clarity and scope of the manuscript. |
|
|
8 |
Some section headings are left in previous page. Arrange them properly |
We sincerely thank the reviewer for this helpful observation. We have reviewed the manuscript and adjusted the formatting to ensure that all section headings are properly aligned with the corresponding content. We greatly appreciate this attention to detail, which contributes to the clarity and presentation of the work. Naturally, we will also take special care during the proofreading stage to ensure that all layout and editorial aspects are finalized to the highest standard. |
All text |
|
9 |
Conclusions could be better projected based on the review conducted |
. We sincerely thank the reviewer for this thoughtful observation, which allowed us to reflect more carefully on how our conclusions communicate the broader significance of the review. We agree that clearly projecting the implications of our findings is essential to reinforcing the value of the work. In this regard, we would like to highlight that a forward-looking reflection—explicitly addressing the public health and implementation perspective of our findings—is already presented in detail in the discussion section titled “A Public Health View on AI in Clinical Practice: Barriers and Facilitators in a Bottom-Up Perspective.” This section was intended to offer a broader interpretive lens on the findings, emphasizing their relevance for future practice, policy, and implementation frameworks. Nevertheless, in response to the reviewer’s valuable input, we have revised the final paragraph of the Conclusions section to ensure stronger alignment with that perspective and to better highlight the projected relevance of our review findings for both clinical and public health contexts. We are sincerely grateful to the reviewer for encouraging these refinements, which have contributed meaningfully to improving the clarity and completeness of the paper. |
Conclusions |
|
10 |
Proofread the manuscript thoroughly |
We sincerely thank the reviewer for this valuable suggestion. Following this recommendation, we have carefully re-read and revised the entire manuscript with particular attention to grammar, syntax, and overall linguistic clarity. To ensure the highest level of quality, the revised version has also been reviewed by a native English speaker with academic experience. We are confident that these improvements contribute to a clearer and more fluent presentation of the work. We are truly grateful for the reviewer’s attention to this important aspect of the manuscript. |
All text |
Thank you again for you time and attention and we hope to have fully addressed your suggestion.
Reviewer 3 Report
Comments and Suggestions for Authors
The manuscript entitled Barriers and Facilitators to Artificial Intelligence Implementation in Diabetes Management from Healthcare Worker’s perspective: A Scoping Review is a critical survey that highlights the role of machine learning in the healthcare sector. However, the quality of the manuscript may be improved further by addressing the following queries:
- In the abstract section, please introduce and expand on the following: PRISMA-ScR, OSF, and JBI tools.
- In the introduction section, please update with the latest article indicating the global status of Diabetes. The present cited reference article is outdated.
- In the introduction section, please split the full-page information into manageable paragraphs for better presentation and readability.
- In the results section, the text in Figure 1 is not readable. The quality of the figure should be improved for better presentation.
- Table 2 can be expanded by discussing studies from the year 2025, as only 7 studies have been discussed herein. It would be better to present diabetes case predictions followed by diabetes-associated symptoms.
- In Table 3, kindly add a footnote for better understanding of the data trends in terms of Low, Medium, Excellent, etc.
- In Figure 2, please endorse the parameters with relevant studies.
- In the conclusion section, authors should provide a future perspective for AI-driven healthcare advancement for further pursuance of this emerging technology.
- References are not written according to the author’s guidelines. For example, please see reference 102 and correct the journal’s full name, Health Expectations, to its abbreviation form.
Author Response
Dear Reviewer, thank you for the effort in revising our manuscript. Please find attached our responses.

Round 2
Reviewer 1 Report
Comments and Suggestions for Authors
In general, the modification was done well. However, it is regrettable that the exclusion criteria itself was deleted and the structure of the discussion is almost the same. In discussion 4.6, one paragraph is too long and the readability is poor, so it is recommended to divide the paragraph into two. Around line 516?
Author Response
Dear Peer,
the answer in the annex file.
Thank you again for your time and efforts.
The Authors

Reviewer 2 Report
Comments and Suggestions for Authors
Dear Authors,
Thank you for addressing all comments.
Author Response
Dear Peer,
thank yu agian for your valuable comments.
Thanks.
The Authors